# Deregulation of New Cell Death Mechanisms in Leukemia

**DOI:** 10.3390/cancers16091657

**Published:** 2024-04-25

**Authors:** Gregorio Favale, Federica Donnarumma, Vincenza Capone, Laura Della Torre, Antonio Beato, Daniela Carannante, Giulia Verrilli, Asmat Nawaz, Francesco Grimaldi, Maria Carla De Simone, Nunzio Del Gaudio, Wouter Leonard Megchelenbrink, Michele Caraglia, Rosaria Benedetti, Lucia Altucci, Vincenzo Carafa

**Affiliations:** 1Dipartimento di Medicina di Precisione, Università degli Studi della Campania “Luigi Vanvitelli”, 80138 Napoli, Italy; gregorio.favale@unicampania.it (G.F.); federica.donnarumma@unicampania.it (F.D.); vincenza.capone@unicampania.it (V.C.); laura.dellatorre@unicampania.it (L.D.T.); antonio.beato@unicampania.it (A.B.); daniela.carannante@unicampania.it (D.C.); giulia.verrilli@unicampania.it (G.V.); asmat.nawaz@unicampania.it (A.N.); nunzio.delgaudio@unicampania.it (N.D.G.); wouterleonard.megchelenbrink@unicampania.it (W.L.M.); michele.caraglia@unicampania.it (M.C.); rosaria.benedetti@unicampania.it (R.B.); lucia.altucci@unicampania.it (L.A.); 2Biogem, Molecular Biology and Genetics Research Institute, 83031 Ariano Irpino, Italy; 3Dipartimento di Medicina Clinica e Chirurgia, Divisione di Ematologia, Università degli Studi di Napoli Federico II, 80131 Napoli, Italy; francesco.grimaldi1@unina.it; 4A.O.R.N. Cardarelli, Divisione di Ematologia, 80131 Napoli, Italy; mariacarla.desimone@aocardarelli.it; 5Princess Máxima Center for Pediatric Oncology, 3584 CS Utrecht, The Netherlands; 6Institute of Experimental Endocrinology and Oncology “Gaetano Salvatore” (IEOS)-National Research Council (CNR), 80131 Napoli, Italy; 7Programma di Epigenetica Medica, A.O.U. “Luigi Vanvitelli”, 80138 Napoli, Italy

**Keywords:** leukemia, regulated cell death, drug resistance, drug discovery, drug-induced cell death

## Abstract

**Simple Summary:**

Resistance to the cell death of neoplastic cells represents one of the main limitations for cancer treatment. The following review describes, in different types of leukemia, the molecular mechanisms driving the new regulated cell death (RCD) processes and their de-regulation. Furthermore, renowned or newly characterized pharmacological strategies, able to modulate the specific mechanisms of RCD, will be addressed.

**Abstract:**

Hematological malignancies are among the top five most frequent forms of cancer in developed countries worldwide. Although the new therapeutic approaches have improved the quality and the life expectancy of patients, the high rate of recurrence and drug resistance are the main issues for counteracting blood disorders. Chemotherapy-resistant leukemic clones activate molecular processes for biological survival, preventing the activation of regulated cell death pathways, leading to cancer progression. In the past decade, leukemia research has predominantly centered around modulating the well-established processes of apoptosis (type I cell death) and autophagy (type II cell death). However, the development of therapy resistance and the adaptive nature of leukemic clones have rendered targeting these cell death pathways ineffective. The identification of novel cell death mechanisms, as categorized by the Nomenclature Committee on Cell Death (NCCD), has provided researchers with new tools to overcome survival mechanisms and activate alternative molecular pathways. This review aims to synthesize information on these recently discovered RCD mechanisms in the major types of leukemia, providing researchers with a comprehensive overview of cell death and its modulation.

## 1. Introduction

Genetic and epigenetic changes are responsible for several abnormalities in the development of hematopoietic stem cells (HSCs). This can occur at any state of differentiation, generating undifferentiated cell clones, characteristic of the leukemogenesis process [1]. Leukemia is defined as the clonal development of leukemic cells in the bone marrow (BM), characterized by an uncontrolled proliferation rate and, in the case of some lymphoid malignancies, the lymphatic tissue. Immature cells are known as “blasts”, and in the BM, under physiological conditions, they represent about 1% of the total cellular population [2]. In acute leukemias, their concentration is greater than 20% in peripheral blood or BM, resulting in a faster symptoms occurrence [3]. In contrast, in chronic leukemias, the concentration of blasts is lower than 20% and symptoms appear gradually [2]. Leukemia is responsible for approximately 2.5% of all new tumor incidence and 3.1% of deaths worldwide; its incidence varies with age, gender, and geographic area [4]. According to the proliferation rate and cell lineage, it is possible to distinguish leukemia into acute myeloid leukemia (AML), chronic myeloid leukemia (CML), acute lymphoblastic leukemia (ALL), and chronic lymphocytic leukemia (CLL) [5,6]. Myelodysplastic syndrome (MDS) is a blood disorder caused by progressive disruption of the normal process of differentiation within the BM, which is associated with a high risk of degeneration into the AML phenotype [7,8]. Evasion of cell death mechanism(s) is a common feature of all malignancies due to the activation of molecular pathways which promote the survival of the neoplastic clone [9], with a consequent resistance to anticancer treatments [10]. During leukemia treatment, the progressive selection of subclones leads to the lack of response to conventional therapies and to the clonal expansion of leukemic stem cells (LSCs), which compete with normal HSCs to gradually replace them in the BM niche [11]. Cell death is an evolutionarily conserved process, with a pivotal role in maintaining cell homeostasis, whose alteration is strictly involved in carcinogenesis. Indeed, the activation of cell death following chemo/radiotherapy treatments underlines the importance of this mechanism in tumor regression [12,13]. It is possible to differentiate accidental cell death (ACD), an unregulated physiological mechanism triggered by unexpected damage, from RCD. The latter is a process occurring under physiological conditions and is recognized as programmed cell death (PCD), occurring without perturbation to homeostasis [12,13]. Adhering to the NCCD guidelines enabled the classification of several types of RCD that may manifest a phenotype which blends characteristics of both apoptotic and necrotic cell death. This classification was based on biological, morphological, and functional attributes [14,15]. Using these parameters, it was feasible to identify intrinsic and extrinsic apoptosis, autophagy-dependent cell death (ADCD), necroptosis, pyroptosis, ferroptosis, parthanatos, NETosis, immunogenic cell death (ICD), entosis, mitochondrial permeability transition (MPT)-driven necrosis, and lysosome-dependent cell death (LDCD) [14,15]. The molecular mechanisms underlying autophagy and apoptosis have been well characterized in both physiological and pathological contexts such as cancer, and have been extensively studied for the treatment of AML, ALL, CML, CLL, and MDS [16,17]. The apoptotic pathway in most tumors, including leukemias, is altered or even blocked because of gene mutations or functional alterations in pro-apoptotic proteins. An in-depth understanding of the molecular processes that govern RCDs, as alternative pathways to the apoptotic one, is fundamental for a better understanding of the neoplastic process, offering a considerable advantage from a therapeutic point of view. Several cell death regulators have been identified as molecular players able to activate and execute their respective RCD pathways, representing potential therapeutic targets (Table 1). The activation of RCD pathways, directly or indirectly, can modulate the effectiveness of CAR-T immunotherapy, which is a therapy approved by the US Food and Drug Administration (FDA) for the treatment of some types of leukemia, ALL, diffuse large B-cell lymphoma (DLBCL), follicular lymphoma (FL), mantle cell lymphoma (ML), and multiple myeloma (MM) [18]. Tumor cells bearing a silenced apoptotic machinery are insensitive to the action of CAR-T. In addition, the molecules released by the execution of the RCD pathways can positively or negatively modulate the effectiveness of the CAR-Ts themselves, as reported for numerous damage-associated molecular patterns (DAMPs) such as high-mobility group box 1 (HMGB1), which, if on the one hand they increase the functionality of the CAR-Ts, on the other hand they sensitize their cytotoxicity linked to the excess release of some pro-inflammatory molecules [19,20]. It is becoming clear that none of these pathways operate alone, but in an intricate crosstalk, sharing different molecular players. The crosstalk between the pathways of apoptosis, necroptosis, and pyroptosis has been identified and has led to the conceptualization of a general cell death phenomenon called PANoptosis, where the formation of a single molecular complex, the PANoptosome, orchestrates the activation of the three RCD pathways [21]. The crosstalk makes it difficult to know when and which pathway will be activated following therapeutic treatment in a condition of inactivation of apoptosis. In this review, we have collected studies connecting the main types of leukemias to the new classified RCD processes, providing an overview of the deregulated death pathways and highlighting the consequent impact on leukemogenesis. We emphasize that among the emerging mechanisms of cell death, inducing necroptosis, ferroptosis, and parthanatos may be regarded as viable alternatives for activating tumor-selective RCD in leukemia. Additionally, targeting specific essential cellular organelles, as seen in LDCD and (MPT)-driven necrosis, also proves to be effective. Moreover, stimulating ICD constitutes a novel immunotherapy approach, guiding the immune system toward the elimination of neoplastic clones. However, pyroptosis acts as a double-edged sword, overcoming resistance to cell death, but also potentially promoting oncogenesis through NLR family pyrin domain containing 3 (NLRP3) inflammasome-mediated inflammation [22]. Maintaining the correct balance of NETosis and neutrophil extracellular traps (NETs) production may regulate the immune response, tipping the scales against tumor cells (Figure 1) [23]. One potential strategy for eradicating neoplastic cells involves re-establishing the regulatory homeostasis of RCD processes, preventing the activation of pro-tumor inflammatory cell deaths and enhancing the tumor-selectivity of drugs.

## 2. Regulation of Necroptosis in Leukemia

Necroptosis is a caspase-independent process regulated by three molecular players, known as receptor-interacting serine/threonine-protein kinases 1 and 3 (RIPK1-RIPK3) and pseudokinase mixed lineage kinase like (MLKL) able to control different cellular responses [24]. Necroptosis, one of the better characterized forms of regulated necrosis [25], displays specific features such as cell enlargement, plasma membrane rupture, transparent cytoplasm, and organelle dilatation, and shares some morphological characteristics with apoptotic cells [26]. The role of necroptosis in cancer is not obvious since, for some type of cancers, it has been implicated in the inhibition of tumorigenesis and metastasis, whereas for others it is linked to increased cancer progression and metastasis. However, this dual role remains to be explored, but undoubtedly the regulation of necroptosis has far-reaching effects both in maintaining cellular health and as a target for disease treatment [27]. Notably, considering that resistance to apoptosis may contribute to leukemogenesis, necroptosis may provide a different approach to overcome treatment resistance by improving drug response. Thus, a growing arsenal of compounds and multiple therapeutic agents have been reported to modulate necroptosis in tumor cells by circumventing acquired or intrinsic apoptosis resistance [27]. The inactivation of RIP1/RIP3 signaling significantly increases the responsiveness of AML cells to interferon-γ (IFN-γ)-induced differentiation. This implies that the inhibition of necroptotic signaling, particularly targeting RIP1 and RIP3, in conjunction with IFN-γ or other differentiation inducers, could be beneficial [28]. Another study demonstrated reduced or absent expression of RIPK3 in human AML primary samples without variations in RIPK1, highlighting the potential of tumor cells to escape necroptosis to survive [29,30]. Interestingly, AML cells carrying the most frequent fms-like tyrosine kinase 3-internal tandem duplication (FLT3-ITD) mutations express robust levels of RIPK1 [31]. ALL cells exhibit a general dysregulation of cell death, e.g., by overexpression of inhibitor of apoptosis (IAP) proteins, which makes them resistant to chemotherapy treatments [32]. In addition, the sensitivity to RIPK1-dependent cell death has previously been demonstrated to represent a particular vulnerability of ALL refractory and recurrent patients, which has remained unexploited by conventional chemotherapy. Notably, RIPK3-dependent necroptosis activation downstream of RIPK1 contributes significantly to antileukemic activity, supporting the identification of RIPK3 as a tumor suppressor [33]. CLL patients exhibit a dysregulated necroptosis mechanism due to RIPK3 and cylindromatosis (CYLD) downregulation, which may explain why malignant B cells accumulate in CLL patients [34]. Several human hematologic malignancies exhibit abnormal expression of lymphoid enhancer-binding factor 1 (LEF1), a crucial transcription factor of the Wnt/ß-catenin pathway and a member of the LEF/T-cell factor (TCF) family [34]. Particularly LEF1 has been found to be a repressor of CYLD. Its high expression is a prognostic factor in CLL and correlates with the low expression of CYLD [35]. Thus, restoring the necroptotic pathway by targeting the LEF1–CYLD axis may provide a novel strategy for the treatment of CLL. However, whether and how RIPK1 is implicated in CLL has not been fully explored yet. By analyzing public transcriptomic data, RIPK1 mRNA levels were observed to be downregulated in some B-cell tumors [36]. Unfortunately, little progress has been made in investigating necroptosis in CML, and likely also for its good prognosis.

## 3. Regulation of Pyroptosis in Leukemia

Pyroptotic cell death was initially characterized as a spontaneous immune mechanism in response to pathogens which occurs as a result of the gasdermine (GSDM)-mediated pore assembly in the plasma membrane followed by the release of the inflammatory cytokines such as interleukin-1 beta (IL-1β) and interleukin-18 (IL-18) into the surrounding microenvironment [37]. Pyroptosis is a process strictly dependent on the activation of the inflammasomes, multiprotein oligomeric complexes with cytosolic localization, whose formation requires the assembly of simple subunits composed of an intracellular receptor/sensor, an adapter protein, and an effector enzyme which initiates the mechanism [38]. Elevated IL-1β levels, coupled with the hyperactivation of the NLRP3 inflammasome, are frequently observed in hematologic malignancies, and exhibit a strong correlation with tumor progression and unfavorable prognosis [22]. Several scientific studies found anomalies and polymorphisms in inflammation-related genes such as *NF-κB*, *NLRP3*, *IL-1β*, and *IL-18*, making them potential prognostic markers for hematological tumors [39,40,41]. The activation of NLRP3 in leukemia is associated with its pro-inflammatory phenotype, yet it seems to be decoupled from the induction of pyroptosis [42]. To induce inflammasomes and pyroptosis in leukemic models, various compounds have been identified that activate intracellular sensors distinct from NLRP3 [43]. Additional studies would be essential to understand how to modulate the pyroptotic process by going deeper into the molecular mechanisms underlying leukemia and the potential therapeutic strategies. In AML, the constitutive activation of KRAS and of the MAPK pathways are crucial events for tumor initiation and progression [44]. The interaction between KRAS and RAC1 increases reactive oxygen species (ROS) production and NLRP3 priming, highlighting a functional link between NLRP3 inflammasome activation and the MAPK pathway [45,46]. The high levels of NLRP3 expression found in AML are also associated with those of the Aryl hydrocarbon receptor (AHR), implicated in immune system regulation, especially in T helper cells (T_h_ cells) subset development [47]. Overexpression of NLRP3 and AHR resulted in an imbalance of Th22 populations at the expense of Th1, leading to impaired differentiation and AML progression [47]. Moreover, various studies have linked the activation of the NLRP3 inflammasome and the production of IL-1β to the deterioration of the structure and function of the BM [48,49]. Because of the basal activation of the NLRP3-inflammasome in AML, which correlates with survival, it has been studied how to activate pyroptosis through the induction of other types of inflammasomes [39,50]. The hyper activation of the NLRP3-inflammasome and the effector caspase 1 is responsible for the drug-resistance processes in ALL patients under glucocorticoid therapy [51]. Indeed, the enzymatic cleavage of the nuclear receptor 3C1 (NR3C1) in the transactivation domain by caspase 1 prevents its nuclear translocation, reducing the sensitivity to glucocorticoids, currently used in therapy [51]. The application of CAR-T therapy, an immunotherapy approach against the ALL cluster of differentiation 19 (CD19^+^) blasts, induces an alternative form of pyroptosis which involves both GSDME/GSDMD together with a cytokine release syndrome [52]. Compared to other types of leukemia, CLL patients display diminished expression levels of NLRP3, coupled with elevated GSDM-E. This combination is strongly associated with an unfavorable prognosis [53]. Despite the limited empirical support, emerging evidence suggests a potential tumor suppressor role for NLRP3 in CLL. This putative function is attributed to its capacity to impede tumorigenesis by suppressing the expression of P2X7R, an adenosine triphosphate (ATP) receptor [54]. Current scientific knowledge does not show a direct link between pyroptosis and CML, but hypothesizes a possible involvement considering the high expression levels of some cytokines involved in the pyroptotic process. Indeed, high levels of IL-1β were found in the CML bone marrow [55], and high levels of interleukin-1 receptor (IL-1R) and interleukin-1 receptor accessory protein (IL-1RAP) were found in CML LSCs [56].

## 4. Regulation of Immunogenic Cell Death in Leukemia

ICD is a distinctive form of RCD that plays a crucial role in activating an immune response against cancer cells in the tumor microenvironment (TME). Unlike conventional cell death processes, ICD induces the release of specific danger signals, known as DAMPs, and immunostimulatory factors. These molecules alert the immune system, triggering the recruitment and activation of antigen-presenting cells (APC) and promoting the subsequent recognition and elimination of cancer cells by cytotoxic T lymphocytes [57]. Key molecular events in ICD involve the exposure of calreticulin (CALR) on the cell surface, the release of ATP, Heat-shock proteins (HSPs), and HMGB1, and the activation of endoplasmic reticulum stress responses [58]. Little evidence is reported in hematological malignancies, including leukemia. Particularly, ICD-related DAMPs, especially CALR and HSPs, may play a crucial role in leukemia by enhancing both innate and adaptive immune responses [59]. Inducing ICD in leukemia cells could render them more vulnerable to the attacks of immune system cells, thereby enhancing the efficacy of immunological therapies and targeted immunotherapies [60]. Although in AML patients the malignant blasts exhibit on the membrane surface several immunostimulatory signals, the activation of antitumor immunity and ICD through natural killer (NK), CD4^+^, and CD8^+^ T cells is closely related to the recognition of the externalized CALR (ecto-CALR) [60]. Specifically, a significant increase in circulating CD4^+^ and CD8^+^ T lymphocytes was found upon recognition of leukemia-associated antigens (LAA) suggesting ecto-CALR and ICD stimulation as a potential strategy to improve clinical patients’ outcomes in AML [60,61]. The formation of the immunosuppressive microenvironment in AML patients is determined by the generation of ATP following chemotherapy treatment by increasing the presence of regulatory T lymphocytes (Tregs) and dendritic cells (DCs) [62]. Moreover, in addition to immune system activation, the action of Tregs and DCs following chemotherapy administration in AML activates the tolerogenic pathways necessary for ICD-related phenomena [62]. Not many studies have been reported about ALL where prognostic predictions for stage III children can be made by assessing the differential expression of genes associated with DAMPs produced within the TME following ICD [63]. Notably, individuals classified in the low-risk group exhibited enrichment in activities related to APCs, NK cells, and T cell activation. This observation implies an inherent antitumor role attributed to ICD in the context of ALL [63].

## 5. Regulation of Ferroptosis in Leukemia

The iron-dependent cell death process, known as ferroptosis, is caused by lipoxygenases (LOXs) lipid peroxidation of polyunsaturated fatty acids (PUFAs) in the plasma membrane [64]. The program includes three primary metabolic processes involving thiols, lipids, and iron by producing lipid peroxidation and, ultimately, cell death [64]. The oxidative mechanisms causing ferroptosis are controlled by the heterodimeric amino acid transporter system X_c_− (sX_c_−), a cystine/glutamate antiporter complex able to regulate the traffic of cystine and glutamate through the cell membrane and the enzymatic activity of glutathione peroxidases (GPXs) [65]. Selected studies conducted on leukemia patients indicates that the excessive intracellular accumulation of iron can significantly augment the sensitivity of leukemia cells to ferroptosis. AML patients show high expression levels of *GPX1*, *GPX3*, *GPX4*, and *GPX7* genes, which probably confer resistance to ferroptosis and appear to be promising biomarkers of poor prognosis [66]. AML cells are protected from oxidative damage by the enzymatic activity of aldehyde dehydrogenase 3a2 (Aldh3a2), which detoxifies the long-chain aliphatic aldehydes produced by lipid peroxidation [67]. Indeed, in AML, knockdown of Aldh3a2 encourages accumulation of toxic metabolites by activating ferroptosis [67]. Several researchers have reported the activation of ferroptosis in AML models, opening new scenarios for the modulation of this process [68]. Poor evidence exists regarding the involvement of ferroptosis in CLL. A comprehensive analysis involving the screening of 110 ferroptotic-related genes in a CLL patient cohort led to the identification of 14 genes with prognostic significance. These genes stratified patients into high-risk and low-risk groups based on their expression profiles and responses to chemotherapy [69]. Additionally, the poorly expressed SLC3A2 receptor in CLL has been observed to exert a modest detoxifying effect via the sX_c_− system [70]. This observation implies potential applications in the development of ferroptosis-inducing compounds targeted against CLL.

## 6. Regulation of NETosis in Leukemia

NETosis is an RCD-targeting neutrophils characterized by the formation of NETs known to play an essential role in innate immune response to infections [71]. NETs are composed of aggregates of decondensed chromatin, granules, and cytoplasmic proteins, which are released into the extracellular space in order to trap and neutralize the pathogen also through the interaction between NETs and the complement cascade components [72]. Tumor-induced NETs act as a scaffold on the tumor cell, delivering pro-tumor mediators which accelerate mitochondrial biogenesis and cell proliferation [73]. Furthermore, tumor-induced NETs seem to be linked to a poor prognosis, elevated thrombotic risk, and systemic inflammation, suggesting that the inhibition of the NETs process can be used as a therapeutic strategy against cancer [23]. In the context of leukemia, abnormal white blood cells, including neutrophils, may experience sustained NETosis activation due to genetic mutations. The release of DNA and cellular components can further aggravate the inflammatory microenvironment associated with leukemia [74]. In the AML subtype, known as acute promyelocytic leukemia (APL), the imbalance of NETs production by neutrophils leads to progressive endothelial damage and vascular leakage [74]. The occurrence of NETosis is linked with the proper function of the neutrophilic cytoskeleton; in contrast, in pathologies showing defects in actin polymerization, such as the rare form of AML known as acute myelocaryoblastic leukemia (AMKL), NETosis cannot happen [75]. In childhood ALL patients, although neutrophil counts are not impaired in the early stages of the disease, it has been demonstrated the low production of NETs before therapy [76]. Following the first remission, a significant increase in NETs production has been observed, suggesting a possible link between the treatment response and the restoration of neutrophil immune capacity [76]. The activation of oxidative reactions and the enzymatic activity of neutrophil elastase (NE) are strongly reduced at the diagnosis of ALL and show a tendency to return to physiological levels after consolidation therapy [77]. One of the best strategies in the treatment of high-risk ALL patients is the allogeneic transplantation of HSCs. Nevertheless, complications such as graft versus host disease (GVHD), can compromise its efficacy [78,79]. Although the role of neutrophils in CLL has traditionally been underestimated, recent studies have brought attention to a notable trend: CLL patients exhibit a higher inclination for NETs release compared to their healthy counterparts. This phenomenon triggers intricate transcriptional mechanisms within leukemic cells, resulting in the upregulation of CD69^+^, CD80^+^, and CD86^+^ markers [80,81]. Furthermore, the existing literature establishes a clear link between elevated NETs and an increased risk of diffuse coagulative strokes [82]. It is plausible that the heightened susceptibility to thrombotic events frequently observed in CLL may, in part, be rooted in dysregulated neutrophil NETosis [83,84]. In CML, it has been demonstrated that the overproduction of NETs is a recurrent event. In addition, the excessive production of NETs induced by the tyrosine kinase inhibitor (TKI) ponatinib probably correlates with increased vascular toxicity [85]. Furthermore, in CML, both neutrophils Ph^+^ e Ph^−^ populations appear to increase the production and the release of NETs in response to platelet-activating factor (PAF) stimulation, suggesting that the increase in NETs does not depend on the presence of the Philadelphia (Ph) chromosome [86]. In the context of leukemia, NETosis is associated with a range of negative impacts, including chronic inflammation, alterations in the microenvironment, and the worsening of symptoms, suggesting the imperative need to inhibit the cell death process to counteract the pathology. Treatment with DNAse-1 significantly reduces the chromatin agglomerates of NETs, opening new approaches for the treatment of hemorrhagic events in APL through the modulation of NETosis [74]. In addition, light chain 3 (LC3) autophagy inhibitors, such as wortmannin and 3-MA, prevent the development of NETs production, indicating the direct link between autophagy activation and NETs formation in neutrophils [87].

## 7. Regulation of Lysosome-Dependent Cell Death in Leukemia

Lysosomes are intracellular organelles responsible for the removal of macromolecules through hydrolytic digestion mediated by acidophilic enzymes, leading to the recycling of many cellular components [88]. When lysosomes are damaged, the lysosomal membrane permeabilization (LMP) can cause the cytoplasmic release of lytic enzymes and activate molecular pathways leading to the so-called LDCD [88]. Despite LDCD being first defined many years ago, the whole process is still not fully understood [89]. The cytoplasmic acidification and the proteolytic action of enzymes such as cathepsins (CTSs), chymotrypsin, and proteases following LMP are the main cause of generalized intracellular damage and of LDCD [90]. Overexpression of lysosomal enzymes e.g., heparanase and CTSs, have been reported in several cancer types such as gastric cancer, colorectal cancer, melanoma, glioma, lung cancer, and leukemia [91,92]. Furthermore, the stabilizing action on the lysosomal membrane carried out by HSP70 was also studied, which can be compromised in cancer causing the increased tendency of lysosomes to LMP [93]. Lysosomal fragility can be harnessed both to selectively eliminate neoplastic clones and to prevent lysosomal drug clearance, thereby limiting the development of drug resistance [94]. Morphological and functional alterations of lysosomes have been reported in AML, but little is known about LDCD [95]. In AML, the high oxidative rate is responsible for the increase in enzymatic activity, lysosomal mass, and membrane damage, which contribute to rise lysosomal fragility without changing their intracellular number [96]. Dysregulation of lysosomes promotes drug clearance and chemoresistance processes, opening the possibility of using lysosome-selective drugs to improve the response to treatments in AML [96]. Hence, inhibiting the production of lysosomal membrane proteins can be used to limit the activation of pro-survival autophagy, causing LDCD and solving the issue of drug resistance in AML [97].

## 8. Regulation of Others RCD in Leukemia

In the landscape of leukemia research, some of the mechanisms in RCD remain relatively unexplored. Limited insights into these pathways have prompted investigations into novel drug interventions designed to activate and modulate these elusive mechanisms. The RCD process known as parthanatos has been disclosed during the study of oxidative damage induced by diabetic hyperglycemia [98]. The pivotal player of the parthanatos molecular mechanism is poly (ADP-ribose)-polymerase-1 (PARP-1), a nuclear enzyme involved in DNA repair and in the regulation of cell division due to its interaction with DNA helicases, topoisomerases, and transcription factors [99]. Many of the roles played by PARP-1 are based on the synthesis of poly (ADP-ribose) (PAR) and their binding to specific targets, including histones on promoters, can modify gene expression and chromatin acetylation levels [100]. Despite a dysregulation of parthanatos being found in several pathological conditions such as retinal diseases, diabetes, cardiovascular diseases, neurological diseases, and solid cancers, limited information is available about its regulation in leukemia [101,102,103]. These are conflicting and currently limited to AML, and there is still much to uncover regarding its regulatory mechanisms, such as metabolic balancing of NAD+ and ATP, alternative activations of PARP-1 and epigenetic targets of PARylation, parthanatos holds the potential to emerge as a possible weapon against hematological malignancies [104].

## 9. RCD Mechanisms in MDS

The evasion of regulated cell death mechanisms in MDS has been inadequately described. RNA-Seq investigation of CD34+ bone marrow cells from MDS or chronic myelomonocytic leukemia (CMML) patients revealed an overexpression of the necroptotic executor MLKL and its relationship with anemia severity. Furthermore, elevated RIPK1 expression supports its role as an inflammatory mediator, classifying it as a predictor of poor overall survival, albeit the mechanism remains unknown [105]. Elevated levels of toll like receptor 4 (TLR4) and TNF receptor-associated factor 6 (TRAF6) have been identified in hematopoietic stem/progenitor cells of MDS patients, contributing to the hyperactivation of the NLRP3 inflammasome via NF-κB [106]. This condition is further exacerbated by mutations in the alarmin S100A9, an immunogenic protein secreted by neutrophils, which instigates an inflammatory response through the TLR4 receptor [107]. Additionally, S100A9 amplifies oxidative stress and activates pyroptosis via NLRP3 in nascent bone marrow cells [106,107]. The overexpression of GSDM-D disrupts the inflammatory homeostasis of the bone marrow in MDS mice, leading to leukocytosis, accelerated aging, and anemia [108]. In the progression of MDS, bone marrow failure and reduced hepatic hepcidin result in inefficient intestinal iron uptake, causing significant iron overload [109]. Elevated oxidative stress in erythroid progenitors is a primary contributor to MDS-associated anemia [110]. However, the onset of cytopenia as a side effect is a potential concern following the activation of ferroptosis in MDS with decitabine (DAC) treatment [111]. Reduced neutrophil microbicidal ability in MDS patients is, in part, attributed to impaired myeloperoxidase (MPO) enzymatic activity, resulting in inadequate ROS release and NETs production [112]. Unfortunately, there is no drug currently that significantly enhances NETs production and activates the adaptive immune system in MDS. Granulocyte colony-stimulating factor (G-CSF) remains the only molecule utilized to reactivate NETs formation in neutrophils, although its clinical efficacy in preventing infections in MDS patients has not been notably improved [113].

## 10. Pharmacological Modulation of Novel RCD Pathways in Leukemia: Unlocking Therapeutic Strategies

Several studies have demonstrated how some RCD pathways can be activated as a response to the action of different drugs, alone or in combination with other treatments. This effect could constitute a crucial strategy to overcome cell death resistance, impacting both prognosis and patients’ outcomes. In the following paragraphs, we provide a summary of RCDs’ activation mediated by different molecules. These are mostly preclinical studies, except for two treatments in combination, capable of activating two specific cell death pathways in vitro, such as pyroptosis and ICD, and are currently being investigated in clinical trials.

### 10.1. Modulation of Necroptosis

Despite current aggressive treatment strategies, the prognosis of AML remains poor due to its low survival and high relapse rate [114]. Moreover, the cell-permeable piperazinyl-quinazolinone compound Erastin, exhibiting lipid peroxidation and oncogene-selective lethality, has been shown to cause mixed cell-type deaths in vitro, including ferroptosis and necroptosis, via the c-Jun-NH(2)-terminal kinase (JNK) and p38 pathway, making AML cells more susceptible to chemotherapy drugs in an RAS-independent manner [115]. A therapeutic breakthrough in AML demonstrated that Birinapant, a synthetic small molecule inhibitor of IAP family proteins, is particularly effective when combined with the clinical caspase 8 inhibitor Emricasan/IDN-6556, promoting necroptosis. Notably, while caspase 8 deletion sensitizes AML cells to Birinapant, the combined loss of caspase 8 and necroptosis effector MLKL prevents Birinapant/IDN-6556-induced death, demonstrating that caspase 8 inhibition sensitizes AML cells to Birinapant-induced necroptosis [116]. A new antagonist of IAP proteins, BV6, classified as a second mitochondria-derived activator of caspases (SMAC) mimetic, offers good therapeutic opportunities [27]. The clinical utility of SMAC mimetic alone and/or in combination with therapy (e.g., epigenetic and chemotherapeutic drugs) indicates that necroptosis may represent, in AML, a different mechanism of cell death that is an alternative to apoptosis [27]. BV6 cooperates with the demethylating agent such as 5-azacytidine (5AC) and DAC to induce cell death in AML following the autocrine/paracrine cycle of tumor necrosis factor α (TNFα) [117]. Interestingly, the BV6/DAC co-treatment bypasses apoptotic resistance switching versus the necroptotic pathway after caspase inhibition [117]. The same effect was found when BV6 acts synergistically with cytarabine (AraC), a key chemotherapy drug used in the treatment of AML [118]. Furthermore, BV6 acts synergistically with histone deacetylase inhibitors (HDACi), such as MS275 and SAHA, to induce necroptotic cell death when caspase activation is inhibited [119]. BV6 also sensitizes FLT3-ITD AML cells toward apoptosis and necroptosis, both alone and in combination with death ligands, for example, CD95L or tumor necrosis factor-related apoptosis-inducing ligand (TRAIL) [31]. Together with SMAC mimetics, inhibition of the transcription factor Homeobox protein/Pre-B-cell leukemia homeobox (HOX/PBX), belonging to the homeobox family, also induces necroptosis, which can be blocked by protein kinase C (PKC)-mediated signaling. Using the short cell-penetrating peptide HXR9, which replicates the conserved hexapeptide in HOX proteins, it is possible to disrupt the connection between HOX and its binding partner PBX, resulting in necroptotic cell death and decreased tumor development [120]. Regardless of resistance to apoptosis, AML cells can be selectively targeted by the GM-CSF diphtheria toxin (DT-GMCSF) leading to apoptosis and caspase-independent necroptosis simultaneously [121]. Granulocyte-macrophage colony-stimulating factor receptor (GM-CSFR) is upregulated in AML and drives tumor growth [122]. According to a mechanism of inhibition of protein synthesis through adenosine diphosphate (ADP)-ribosylation of eukaryotic translation elongation factor-2 (eEF-2), treatment with diphtheria toxin alone is sufficient to simultaneously activate the cell death pathways of apoptosis and necroptosis in leukemic cells compared to normal hematopoietic cells [121]. Although this treatment has so far demonstrated good clinical effectiveness, other fusion toxin drugs, e.g., DT-IL3, are currently undergoing clinical trials to treat AML to mitigate the toxicity [121]. As in AML, small molecule SMAC mimetics (i.e., BV6, LCL161, and Birinapant) and demethylating agents synergize to induce cell death via both the apoptotic and necroptotic pathways [123]. BV6 and 5AC cooperate to trigger caspase-dependent cell death, but when caspase activation is blocked, the co-treatment results in caspase-independent non-apoptotic cell death. This indicates a switch from apoptosis to necroptosis, which represents an alternative approach for the treatment of ALL [123]. In ALL cells lacking FAS-associated death domain protein (FADD) and caspase 8, and being refractory to apoptosis, BV6 and TNFα have been able to activate the necroptosis pathway [124,125]. Dexamethasone (DEXA) and BV6 together can trigger necroptotic cell death in ALL cells that express the RIPK3 protein but lack caspase activation due to low caspase 8 expression or its pharmacological suppression [126]. The co-treatment promotes the loss of mitochondrial membrane potential (MMP), ROS generation, BCL2 homologous antagonist/killer (BAK) activation, and interruption of mitochondrial respiration [126]. Proteasome inhibition has shown potential as an anticancer drug, but failure of the ubiquitin-proteasome system (UPS) can result in several diseases, including carcinogenesis [127]. When RIPK3 kinase activity is intact and MLKL is available, the proteasome inhibitors MG132 and Bortezomib can trigger RIPK3-dependent necroptosis in Jurkat T-cell leukemia cell lines without inhibiting caspase 8. However, when MLKL recruitment to RIPK3 is blocked, RIPK3 binds RIPK1, FADD, and caspase 8 to create a complex that triggers apoptosis [128]. For the treatment of patients with chronic or accelerated CML who are resistant or intolerant to TKIs, the natural alkaloid Homoharringtonine (HHT) induces a distinct pathway of necroptosis mediated by TRAIL via the RIPK1/RIPK3/MLKL signaling [129]. HHT was recently approved by the FDA as an alternative to cycloheximide (CHX)-mediated necroptosis, which is cytotoxic and unsuitable for the treatment of cancer patients [129]. LQFM018, a piperazine-containing drug, induces CML cell death in vitro activating necroptotic signaling, probably with the involvement of the dopamine D4 receptor [67]. LQFM018 triggers the characteristic events of necroptosis, including the mitochondrial damage followed by ROS production and the increase in TNF-R1 protein and CYLD mRNA expression levels, without involving caspase 3 and 8 NF-ĸB activation [130]. Since these results also revealed low toxicity in mice in vivo, further studies are needed to better define the mechanisms involved in its antitumor effects. For the treatment of CLL, restoring the necroptotic pathway by targeting the LEF1-CYLD axis may provide a new therapeutic strategy. Ethacrynic acid (EA), a diuretic drug that is a particular antagonist of Wnt signaling, has been shown to specifically kill CLL cells by interfering with the binding of LEF1 to DNA and restoring CYLD expression [131]. Furthermore, it appears that elevated levels of CXCL-1 in CLL cells regulate LEF1 expression, resulting in dysregulated necroptosis. Together with TNF- α and the pan-caspase inhibitor z-VAD-fmk (z-VAD), selenite can block the expression of CXCL1 and repair the defective necropotonic pathway of CLL cells (Table 2) [132].

### 10.2. Modulation of Pyroptosis

Several scientific papers have demonstrated how some RCD pathways can be activated as a response to the action of different drugs, alone or in combination with other treatments. This effect could constitute a crucial strategy to overcome cell death resistance, impacting both prognosis and patients’ outcomes. Combined treatment between Dasatinib and IFN-α activates the pyroptotic pathway and is in phase IV of the clinical trial, conducted to expand treatment options of patients with Ph+ ALL during long-term maintenance therapy [133]. The study investigated the effectiveness of the combination therapy, demonstrating good tolerability of the drugs and a better response of patients to this treatment, compared to those ones treated with the TKI alone (chiCTR1800015763) [133]. The inhibition of the IL-1β pathway by Anakinra, an antagonist of the IL-1R1 or through the monoclonal IL1RAP antibody mAb81.2, attenuated the expansion of CML LSCs [134]. Moreover, treatment with the inhibitor of the dipeptidyl peptidases DPP8/DPP9, Val-BoroPro, induces pyroptosis through the activation of the NLRP1 sensor in AML [135]. The pyroptotic pathway can be also activated after the treatment with natural compounds such as curcumin and Ardisianone. Curcumin, an active molecule extracted from turmeric, can simultaneously activate the NLRC4, AIM2, and IFI16 inflammasomes, promoting the activation of caspase 1, the GSDM-D cleavage, and pyroptotic cell death [136]. Ardisianone, a natural benzoquinone extracted from the roots and stems of Ardisia virens, activates pyroptosis through the negative modulation of IAPs and the caspase-1-mediated GSDM-D cleavage, allowing for monocytic differentiation towards a macrophagic phenotype (Table 3) [137].

### 10.3. Modulation of ICD

Chemotherapy drugs, such as Anthracyclines or ionizing irradiation, can trigger the ICD through the modulation of ICD-related DAMPs, suggesting that their action could be fundamental during cancer development [59,138,139]. Some in vitro and in vivo studies reported that, after chemotherapy treatment, DCs expressing indoleamine 2,3-dioxygenase 1 (IDO1) and Tregs participate in the creation of an immunosuppressive environment, and the balance between tolerance and immunity activation could lead to several final immune system responses [140]. In primary AML blasts and the HL-60 leukemic cell line, treatments with Etoposide (ETO) and Daunorubicin (DNR) can induce ICD mechanisms through the exposure on the cell surface of both CALR and HSPs and also induce the release of HMGB1 and ATP [141]. For consolidation therapy in patients with acute myeloid leukemia in first remission, the effectiveness of the administration of anthracycline before immunotherapy with histamine dihydrochloride (HDC) and IL-2 has been demonstrated. This study is currently in phase IV of the clinical trial. This improvement is associated with increased blood levels of CD8^+^ T_EM_ cells, potentially due to enhanced exposure of CALR and the activation of ICD in leukemic cells (NCT01347996) [142]. CALR is not only exposed via exocytosis on the cell membrane after anthracycline treatment, but it is also present in the serum of AML patients after induction therapy [143]. Moreover, the exposure of CALR in primary human AML cells was detected in 65% of patients after the treatment with all-trans retinoic acid (ATRA) [144]. In this context, it was also demonstrated that for both mice and AML patients, CALR exposure on malignant blasts stimulates anticancer immunity through type I IFNs overproduction, which activates the cytotoxic action of T and NK cells inducing the innate and adaptative immune responses [61]. Furthermore, the activation of the ICD in ALL has been demonstrated following treatment with CM-272, a potent reversible inhibitor of both G9a and DNA methyltransferases (DNMTs) [145]. The use of CM-272 has been shown to improve the in vivo and in vitro survival of hematologic ALL models by activating the type 1 IFN response leading to the expression of IFN-stimulated genes (ISGs) as well as the induction of ICD [145]. Since the transmembrane protein CD47 is overexpressed in many types of hematologic cancers and especially in ALL, where it is able to drive cell death processes through its binding to the signal regulatory protein α (SIRPα) or thrombospondin-1 (TSP1), the use of the first-described serum-stable CD47-agonist peptide PKBH1 has been shown to induce ICD in ALL [146]. Indeed, PKBH1 has been shown to induce ICD in T-ALL leukemia cell lines through the expression of CALR on the cell surface and by releasing HSPs as well as HMGB1, shedding light on the therapeutic potential of CD47 agonist peptides in the treatment of ALL (Table 4) [146].

### 10.4. Modulation of Ferroptosis

Numerous compounds have been identified, by the scientific literature, which have the ability to modulate ferroptosis, acting either as inhibitors or as inducers. Notably, iron chelators like deferoxamine (DFO) or deferiprone (Def) have been recognized for their efficacy in suppressing ferroptosis processes. Conversely, external agents such as ferric ammonium citrate (FAC) or Erastin have demonstrated their capacity to elicit and promote ferroptosis [147,148,149]. Erastin, known as the inhibitor of the sXc−, can enhance ferroptosis in HL-60 cells through the activation of the JNK/p38 molecular pathway, but not ERK, increasing the nuclear translocation of HMGB1, and also supporting the therapeutic effect of AraC and doxorubicin (DXR) [115,150]. The transcriptional activator of p53, APR-246, alone and in combination with the GPX4 inhibitors RSL3 and FINO2, reduces the ability of AML clones to detoxify membrane lipid peroxidation, activating ferroptosis [151]. Ferroptosis can also be regulated by the activity of long non-coding RNAs (lncRNAs), some of which can be considered good prognostic markers in AML [152]. A recent characterized ncRNA, known as CircKDM4C, acts as a microRNA (miRNA) sponge on hsa-let-7b-5p miRNA, exerting a tumor-suppressive role by upregulating p53 and inducing ferroptosis [153]. Furthermore, in AML, the activation of the autophagic mechanism and the subsequent induction of ferroptosis is well demonstrated [154]. Dihydroartemisinin (DHA) and Typaneoside (TYP) have shown the ability to activate the progressive autophagy-dependent ferritin degradation via the AMP-activated protein kinase (AMPK)/mammalian target of rapamycin (mTOR) pathway, increasing oxidative damage and causing ferroptosis [155,156]. By examining the effects of an ATRA-derived compound ATPR on AML models, it has been highlighted that autophagy is an upstream process of ferroptosis activation, induced by nuclear factor erythroid 2–related factor 2 (NRF-2) downregulation and ROS production [157]. The inhibition of the autophagic mechanism upon 3-methyladenine (3-MA) treatment significantly reduces the ferroptosis caused by the TKI neratinib [158]. In ALL, the combination between the SMAC mimetic BV6 and the GPX4 inhibitor RSL3 activates ferroptosis through the modulation of the redox equilibrium [159]. In addition, RSL3 can overcome resistance to apoptotic cell death in a FADD-deficient ALL model through the activation of ferroptosis, which can be prevented counteracting PUFAs peroxidation with the pan-LOX inhibitor nordihydroguaiaretic acid (NDGA) [160]. Ferroptosis not only occurs as a unique death mechanism, but can coexist with other RCD processes. The well-known antimalaria drug Artesunate (ART) exhibited strong cytotoxic and oxidative effects in ALL by activating ferroptosis in conjunction with apoptosis and necroptosis [161]. The mushroom-derived component poricoic acid (PAA) triggers, in T-ALL cells, the activation of the autophagy process via the AMPK/mTOR/ Microtubule-associated protein 1A/1B-LC3, and in secondary ferroptosis via the GSH depletion and GPX4 inhibition [162]. In addition, it is reported that the autophagy activator rapamycin (RAPA) can sensitize ALL cells to Erastin, activating ferroptosis [163]. Increasing scientific evidence associates high levels of circular RNAs (circRNAs) in the bone marrow with the development and evolution of leukemias [164]. The knockdown of the circRNA circ_0000745 in ALL patients and in leukemia cellular models leads to cell cycle arrest, affecting the glycolytic mechanism and activating ferroptosis [165]. Thioredoxin reductase 1 (TXNRD1), a flavoenzyme containing selenocysteine, plays a crucial role in protecting against ferroptosis-induced oxidative damage. Inhibition of TXNRD1, achieved with the TXNRD1-inhibitor auranofin, has been shown to induce ferroptosis in CML models [166]. Conversely, ferroptosis induced using IM therapy in CML patients leads to adverse effects, such as cardiotoxicity, due to reduced GPX4 levels, increased p53 expression, and elevated production of ROS in cardiomyocytes. This cascade results in elevated serum concentrations of creatine kinase (CK) and lactate dehydrogenase (LDH) [167,168]. Due to the iron dependence of ferroptosis, treatment with DAC has been shown to reduce the antioxidant activity of GPX4 in MDS cells, leading to increased ROS release and the evolution of AML through ferroptosis activation [111]. This corroborates earlier findings in both low- and high-risk MDS patients (Table 5).

### 10.5. Modulation of LDCD

One of the best strategies to modulate lysosomal function and activate LDCD processes in tumor cells is to act on channel proteins responsible of the ion transport to and from the lumen of lysosomes [169]. Two-pore channel 2 (TPC2) is a cationic pore responsible for regulating sodium and calcium levels as well as lysosomal pH, known to be highly expressed in ALL [170,171]. Knockout or inhibition of TPC2 in ALL patients and cellular models increases lysosomal pH and enhances the response to the cytostatic drugs vincristine (VCR), DXR, and topotecan (TPT), activating CTSB-mediated LDCD [172]. A key component of standard treatments for ALL involves the use of BCL-2 inhibitors. However, the development of mutations can lead to therapy resistance [173,174]. The lysosomotropic drug derived from betulinic acid, B10, has been shown to enhance the effects of the BCL-2 inhibitor ABT-263, increasing lysosomal permeabilization and activating LDCD in different ALL models [175]. The precise mechanism underlying the induction of LMP by lysosomotropic drugs, such as the sigma receptor agonist siramesine, and its subsequent impact on mitochondrial membrane potential, leading to LDCD, remains elusive. However, it has been demonstrated that the selective killing of primary CLL cells can be achieved by impairing the sphingolipid metabolism [176]. The use of monoclonal antibodies such as anti CD20 and HLA-DR monoclonal antibodies triggers, in primary CLL samples, a rapid LDCD with cytoplasmic release of CTSB without the involvement of caspases [177]. The antimalarial drug tafenoquine (TQ) rapidly enters the lysosomes of CLL cancer cells causing LMP, the subsequent peroxidation of lipids, and the simultaneous activation of LDCD and mitochondrial apoptosis [178]. The impact of LDCD has not been extensively characterized in CML, although alterations of lysosomal functions are known in this cancer disease [179]. IM acts in the cytoplasm of the neoplastic cell by preventing the autophosphorylation of the ABL/BCR fusion protein [180]. The ATP-binding cassette subfamily A member 3 (ABCA3) transporter is in the lysosomal membrane of CML cells, and its expression is directly related to the sequestration of IM in lysosomes and drug resistance [181]. Imatinib resistance can be overcome by combining IM with mefloquine (MQ), suggesting lysosome targeting as a cross-sectional pathway to induce cell death in TKIs-resistant CML cells [182]. Additionally, MQ can trigger apoptosis secondary to LDCD with the release of CTSs and the increase of oxidative damage [183]. The natural compound extracted from plants of the solenaceae family solamargine (SM) is a steroid alkaloid glycoside and shows cytotoxic effects in various tumor models [184]. Specifically, in a CML model, SM activates apoptosis following lysosomal swelling, cytoplasmic activation of CTSB, and mitochondrial damage [179]. This evidence supports the role of LCDD as an early phenomenon of intrinsic apoptosis in CML (Table 6).

### 10.6. Modulation of Other RCD Mechanisms

The immunodeficient status of leukemia patients affected by AML could be related to the ability of malignant ROS-producing myeloid clones to activate parthanatos through the ERK1/2/PARP-1 axis in NK lymphocyte populations [185,186]. To recover lymphocyte function and reverse the parthanatos mechanism, new immunotherapy strategies have been carried out using ROS formation inhibitors or ERK1/2 inhibitors [187,188,189]. With regard to hematological malignancies, parthanatos activation has been reported in 50% of AML patients undergoing AraC and anthracycline therapy, while resistant patients show low expression levels of PARP-1, classifying it as a potential biomarker of drug resistance [104]. In addition, AraC activates parthanatos in OCI-AML3 cells, causing PAR accumulation and cytoplasmic translocation of the apoptosis-inducing factor (AIF) [104]. Different drugs promote parthanatos cell death in AML, such as the naphthoquinone-derived drug known in traditional Chinese medicine as “shikonin”, which has shown the ability to simultaneously induce apoptosis, necroptosis, and parthanatos via γH2AX hyper-activation of PARP-1 [190]. Furthermore, the small molecule APO866, an inhibitor of nicotinamide phosphoribosyltransferase (NAMPT) and NAD^+^ biosynthesis, can induce parthanatos in AML models by NAD^+^ depletion, mitochondrial polarization, and PARP-1 activation, alone and in combination with ETO [191]. Furthermore, a study on AML reported the selective and potent anti-leukemic effects induced by the aminoalkylindole derivative WIN-55,212-2 (WIN-55), which acts as a parthanatos inductor promoting nuclear translocation of AIF with a glycolytic metabolic drop and hyperactivation of PARP-1. This effect is completely reversed by using the PARP-1 inhibitor Olaparib [192]. The plant-derived flavonoid resveratrol has been shown to induce the potential mitochondrial membrane imbalance in ALL patients, and its combination with the mitochondrial permeability transition pores (mPTPs) inhibitor cyclosporine A (CsA) potentiates the cytotoxic effect of the natural compound in the Jurkat cell line [193,194]. CsA prevents the opening of mPTPs through the sequestration of Ca^2+^ ions and the inhibition of CypD, leading to apoptosis activation instead of necrosis [194]. Furthermore, in ALL cellular models, it has been shown that cannabidiol (CBD) triggers certain processes typical of apoptosis, such as cytosolic migration of Cytc and caspase 9 activation, but causes cell death by MPT-driven necrosis by completely arresting oxidative phosphorylation and mitochondrial production of ATP [195]. Regarding leukemia, NETotic cell death is associated with a range of negative impacts, including chronic inflammation, alterations in the microenvironment, and a worsening of symptoms, suggesting the imperative need to inhibit the process to counteract the pathology. Treatment with DNAse-1 significantly reduces the chromatin agglomerates of NETs, opening new approaches for the treatment of hemorrhagic events in APL through the modulation of NETosis [74]. In addition, LC3 autophagy inhibitors, such as wortmannin and 3-MA, prevent the development of NETs production, indicating the direct link between autophagy activation and NETs formation in neutrophils [87]. MPT is a biological event characterized by the modification of the potential and permeability of the mitochondrial membranes [196]. MPT is activated by elevated levels of mitochondrial Ca^2+^ and ROS, whilst being inhibited by Mg^2+^ and adenosine diphosphate (ADP) [197]. The complete loss of membrane potential and cessation of ATP synthesis are distinctive characteristics of MPT-driven necrosis [198]. Currently, there is a lack of direct information linking MPT-driven necrosis regulation and leukemia. However, the literature suggests the intriguing possibility of activating this mechanism using specific drugs. CBD, a cannabis sativa-derivative compound, which has been shown to trigger apoptosis in leukemic systems, has been researched in the past for its potential anticancer properties [199]. More recently, in ALL cellular models, it has been shown that CBD triggers certain processes typical of apoptosis, such as cytosolic migration of Cytc and caspase 9 activation, but causes cell death by MPT-driven necrosis by completely arresting oxidative phosphorylation and the mitochondrial production of ATP (Table 7) [195].

## 11. Conclusions

The identification of new cell death processes has expanded and strengthened knowledge about the role of cell death dysregulation as a fundamental event for leukemogenesis. While advancements in leukemia treatment have significantly improved patient outcomes, several anomalies in the apoptotic pathways are responsible for the uncontrolled proliferation of leukemic cells, and their resistance to pharmacological treatments. Addressing these limitations is crucial for enhancing therapeutic efficacy and patient prognosis. The identification and the molecular characterization of the role of a single player involved in the activation of one or more RCDs, through a crosstalk of action, has broadened the therapeutic perspectives through the identification of new molecular targets. The characterization of RCDs has led to the functional identification of different drugs capable of modulating them. Among these drugs, two are in phase IV of the clinical trial. For some RCDs such as necroptosis, pyroptosis, or ferroptosis, several drugs are identified, increasing the attractiveness of these molecular pathways as alternatives to apoptotic blockade. The role of RCDs as possible therapeutic targets represents one of the most discussed topics to date. The exploration of novel cellular RCD mechanisms provides a promising strategy for overcoming resistance mechanisms and identifying innovative alternatives in leukemia management. The crosstalk between the different pathways and the coexistence of some of them that can operate simultaneously make the study of cell death a complex scenario. By deepening the understanding of how these cell death mechanisms can be activated, solid foundations can be laid for the development of targeted interventions that can potentially revolutionize leukemia treatment strategies, offering renewed “weapons” to counteract the survival of neoplastic clones, which is necessary to discover new ways to treat hematological malignancies.

## Figures and Tables

**Figure 1 cancers-16-01657-f001:**
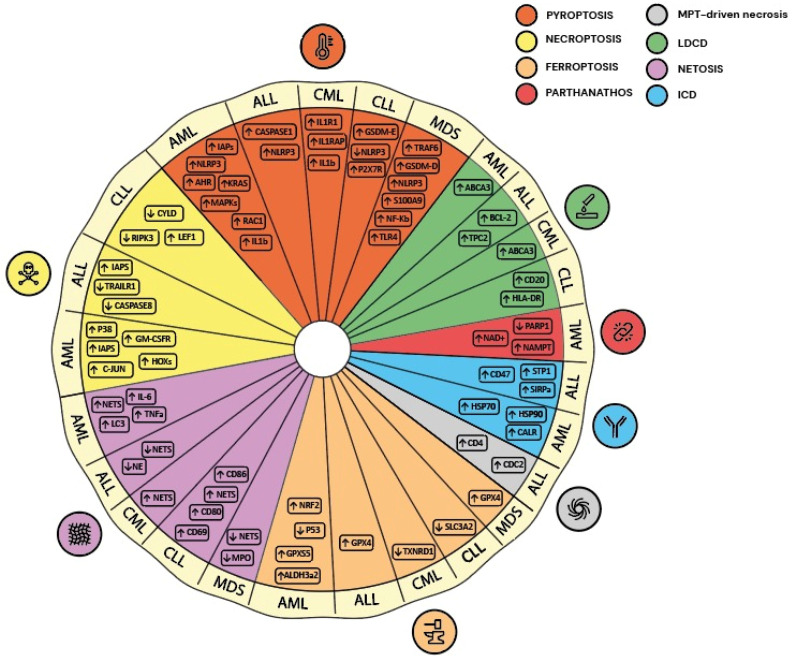
Schematic representation of key up- and down-regulated players of the RCD pathways in different leukemia subtypes.

**Table 1 cancers-16-01657-t001:** Summary of the key biomarkers for identifying the potential targets for overcoming cell death resistance in leukemia cells.

RCD	POTENTIAL DRUG TARGETS
Necroptosis	RIPK1, RIPK3, MLKL, TNFR1, JNK, p38, CYLD
Pyroptosis	NLRP1, NLRC4, AIM2, IFI16, caspase 1, GSDMD, HMGB1
ICD	CALR, HSP70, HSP90, HMGB1, P2X7, ATP, PD1
Ferroptosis	GPX4, Ferritin, GSH, LOXs, cysteine, NRF-2, JNK, p38, sX_c_−, Aldh3a2, TXNRD1
LDCD	TPC2, CTSB, CD20, ABCA3
Parthanatos	ERK1/2, PARP-1, AIF, γH2AX, NAMPT
MPT-driven necrosis	CBRs

**Table 2 cancers-16-01657-t002:** Overview of the drugs inducing necroptosis and the related pathways involved in each reported leukemia subtype.

Disease	Drug	Target	Mechanism of Necroptosis	Ref.
AML	Birinapant + Emricasan	cIAPs, Caspase 8	TNFR1 signaling, RIPK1/RIPK3/ MLKL dependent	[116]
	BV6 + 5AC or DAC	cIAPs, DNA methylation	RIPK1/RIPK3/ MLKL dependent, autocrine TNFα	[117]
	BV6 + AraC	cIAPs, DNA synthesis	RIPK1/RIPK3/MLKL dependent, autocrine TNFα	[118]
	BV6 + MS275 BV6 + SAHA	cIAPs, HDACs	RIPK1/RIPK3/MLKL dependent, autocrine TNFα	[119]
	BV6 + DAMPs	RIPK1	Caspase-dependent and independent cell death	[31]
	HXR9	HOX/PBX dimers	RIPK1 dependent	[120]
	DT-GMCSF	Inhibition of protein synthesis	RIPK1 dependent	[121]
	Erastin	Unknown	RIPK3 dependent; c-JNK and p38 dependent	[115]
ALL	BV6 + DEXA	cIAPs, Glucocorticoid receptor	RIPK3/MLKL dependent, BAK activation	[126]
	BV6, LCL161, Birinapant + 5AC or DAC	cIAPs, DNA methylation	RIPK1/RIPK3/ MLKL dependent, autocrine TNFα	[123]
	MG132, Bortezomib	Proteasome	RIPK3/MLKL dependent	[128]
CML	CHX, HHT	Unknown	TRAILR/RIPK1/RIPK3/MLKL	[129]
	LQFM018	Unknown	TNFR1 and CYLD upregulation	[130]
CLL	EA	LEF1	CYLD activation	[131]
	sodium selenite + TNF-α + z-VAD	CXCL-1	RIPK1/RIPK3/MLKL dependent	[132]

**Table 3 cancers-16-01657-t003:** Overview of the drugs inducing pyroptosis and the related pathways involved in each reported leukemia subtype.

Disease	Drug	Target	Mechanism of Pyroptosis	Ref.
AML	Val-BoroPro	DPP8/DPP9	NLRP1 mediated caspase 1 activation	[135]
	Curcumin	ISG3	Caspase 1 mediated GSDM-D cleavage	[136]
	Ardisianone	IAPs, TNFR2	Caspase 1 mediated GSDM-D cleavage	[137]
ALL	DAS + IFN-α Phase IV clinical trial (chiCTR1800015763)	ABL, Src	Caspase 1 mediated GSDM-D cleavage and IL-1β release	[133]

**Table 4 cancers-16-01657-t004:** Overview of drugs inducing ICD and the related pathways involved in each reported leukemia subtype.

Disease	Drug	Target	Mechanism of ICD	Ref.
AML	ETO, DNR	DNA topoisomerase II; DNA/RNA synthesis	CALR and HSPs surface exposure, HMGB1 and ATP release	[141]
	DNR + AraC	DNA/RNA synthesis	Increased expression of IDO1; Tregs-dependent regulation	[140]
	anthracyclines + HDC/IL-2 Phase IV clinical trial (NCT01347996)	DNA/RNA synthesis	CALR exposure and HSP70 release	[142]
	Anthracyclines	DNA/RNA synthesis; Topoisomerase II; nuclear and cytoplasmatic sites	CALR exposure and release	[143]
	ATRA	PML/RARα	CALR exposure	[144]
ALL	CM-272	G9a and DNMTs	Type 1 IFN response and ISGs activation	[145]
	PKBH1	CD47	CALR exposure, HSPs and HMGB1 release	[146]

**Table 5 cancers-16-01657-t005:** Overview of the drugs inducing ferroptosis and the related pathways involved in each reported leukemia subtype.

Disease	Drug	Target	Mechanism of Ferroptosis	Ref.
AML	Erastin	Unknown	GPX4 inhibition and c-JNK/p38 activation	[115]
	DHA, TYP	NDUFS3, SDHB, UQCRFS1	Ferritin degradation via AMPK/mTOR and ROS production, redox instability	[155,156]
	ATPR	NRF-2	NRF-2 downregulation and ROS production, redox instability	[157]
ALL	RSL3 + BV6	GPX4	Iron-dependent PUFAs peroxidation by LOXs	[159]
	RSL3	GPX4	Iron-dependent PUFAs peroxidation by LOXs	[160]
	ART	DNA/RNA synthesis	Iron-dependent ROS production	[161]
	PAA	Unknown	GPX4 inhibition, GSH depletion, and ROS production	[162]
	Erastin + RAPA	Unknown	FBXW7 downregulation, VDAC3 upregulation	[163]
CML	Auranofin	TXNRD1	TXNRD1 downregulation, cysteine depletion, and ROS production	[166]
MDS	DAC	DNA synthesis	GSH depletion, GPX4 inactivation, and ROS production	[111]

**Table 6 cancers-16-01657-t006:** Overview of the drugs inducing LDCD and the related pathways involved in each reported leukemia subtype.

Disease	Drug	Target	Mechanism of LDCD	Ref.
AML	MQ	Protein synthesis	Oxidative damage, lysosomal permeabilization and CTPs cytosol release	[183]
ALL	VCR, DXR, TPT + Narigenin/ Tetrandrine	Microtubule polymerization, DNA synthesis, TPC2	Increase in lysosomal PH, lysosomal permeabilization and CTSB cytosol release	[172]
	ABT-263 + B10	BCL-2	Lysosomal permeabilization, loss of mitochondrial membrane potential	[175]
CML	SM	Unknown	Lysosomal permeabilization, CTSB release, mitochondrial damage	[179]
CLL	Siramesine	Sigma receptor	Lipid peroxidation and LMP	[176]
	Tositumomab, L243	CD20, HLA-DR	Lysosomal membrane permeabilization and CTSB cytosol release	[177]

**Table 7 cancers-16-01657-t007:** Overview on drugs-inducing parthanatos and MPT-driven necrosis and related pathways involved in each reported leukemia subtype.

Cell Death	Disease	Drug	Target	Molecular Mechanism	Ref.
Parthanatos	AML	Ara-C	DNA synthesis	PAR accumulation and AIF nuclear translocation	[104]
		Shikonin	PMK2	γH2AX phosphorylation and PARP-1 activation	[190]
		APO-866	NAMPT	NAD^+^ and ATP depletion, AIF nuclear translocation and PARP-1 activation	[191]
		WIN-55	CBRs	AIF nuclear translocation and PARP-1 activation, glycolytic metabolic drop	[192]
MPT-driven necrosis	ALL	CBD	CBRs	Ca^2+^ overload, MPT formation, oxidative phosphorylation arrest	[195]

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
