# Peer review of "Deregulation of New Cell Death Mechanisms in Leukemia"

_cancers, 2024, doi:10.3390/cancers16091657_

Round 1

Reviewer 1 Report

Comments and Suggestions for Authors

The review is well written and gives a lot of informations about regulated cell death in hematological malignancies as well as the ways to  modulate RCD in leukemias. 

To improve the review, 

1. The authors should specify in the table if the pharmacological strategy has been tested in clinical or in preclinical models. If there is clinical studies published or presented in meetings, the author should indicate the phase of the trials and the references. It would help the clinicians to have a quick look at the drugs having the potential to overcome apoptosis in the clinics and that could be used in future clinical trials for venetoclax resistant AMLs for instance.

2. The authors could emphasize the cell death mechanisms that are triggered by immunotherapies and that can be used to increase the efficacy of immunotherapies, in particular CAR-t cells. This aspect could help the clinicians to set up new clinical trials for increase CAR-t cell efficacy by using drugs modulating alternative cell deaths.

3. Finally, in the authors could give their opinion about the best way to overcome apoptosis in hematological malignancies. According to them, what would be the best cell death mechanism(s) to trigger to treat resistant diseases?

Author Response

Reply to comments and suggestions for Authors

To the Editor of Cancers,

We greatly thank Editor and Referees for their interest in our manuscript entitled “Deregulation of new cell death mechanisms in Leukemia” and for reviewing it. We have carried out the appropriate changes suggested by reviewer implementing the text and tables accordingly.

Below there is a point-by-point response to referee’s questions, reporting our answers in red colour. 

We hope that the answers are satisfactory and the improvement of the manuscript makes it acceptable for publication.

Reviewer Comments:

Reviewer1 

To improve the review, 

Q1. The authors should specify in the table if the pharmacological strategy has been tested in clinical or in preclinical models. If there is clinical studies published or presented in meetings, the author should indicate the phase of the trials and the references. It would help the clinicians to have a quick look at the drugs having the potential to overcome apoptosis in the clinics and that could be used in future clinical trials for venetoclax resistant AMLs for instance.

A1. Thanks a lot for your useful comment. Among all pharmacological strategies described, only for two of the drugs are reported clinical studies. We added a short introduction (page 10, lines 409-413), followed by the reported clinical trials. These studies are added in the respective paragraphs and highlighted in the table (page 13, lines 503-508) (pages 13-14, lines 535-540). The corresponding tables are also adapted (page 13, Table 3) (page 14, Table 4).

Q2. The authors could emphasize the cell death mechanisms that are triggered by immunotherapies and that can be used to increase the efficacy of immunotherapies, in particular CAR-t cells. This aspect could help the clinicians to set up new clinical trials for increase CAR-t cell efficacy by using drugs modulating alternative cell deaths.

A2. Thanks a lot for this useful comment. As you pointed out, regulated cell death mechanisms can implement the efficacy of immunotherapy, with particular reference to CAR-Ts. Specifically, some molecules released upon activation of cell death pathways can increase or decrease the efficacy of CAR-Ts. Hence, the need for more in-depth studies on the molecular mechanisms underlying the process of RCDs and the role of related molecular interactors. The text has been modified as requested and is reported in the introduction section. (page 3, lines 79-106).

Q3. Finally, in the authors could give their opinion about the best way to overcome apoptosis in hematological malignancies. According to them, what would be the best cell death mechanism(s) to trigger to treat resistant diseases?

A3. Thank you for this comment. The characterization of new mechanisms of regulated death offers significant therapeutic advantages; where is present a blockage of apoptotic mechanisms, the activation of alternative pathways can solve the problem of resistance to cell death. To date, there is no specific evidence regarding the death pathway preferentially activated as a result of the block of apoptosis. Clearly for some pathways (e.g. necroptosis, pyroptosis and ferroptosis), more studied and characterized, there is clearer information regarding the mechanism of action and the different molecular actors activated. This could lead to the hypothesis that these could act as preferential pathways in the event of blocking apoptosis, and thus overcome the phenomena of resistance to cell death. Our opinion on this matter is that there is no “best way to overcome apoptosis”, but the identification of PANoptosis suggests that many pathways, not just the three described, can coexist together. Therefore, based on molecular availability it will be possible to activate one pathway in place of another. To be able to answer this with certainty, the complete characterization of the RCD processes is necessary, which will clarify their role in tumorigenesis and therefore in leukemogenesis, providing useful details about their activation. (Page 3, lines 97-106 and conclusion section)

Reviewer 2 Report

Comments and Suggestions for Authors

This review summarized the regulated cell death (RCD) mechanisms in the major types of leukemia, providing researchers with a comprehensive overview of cell death and its modulation. The topic fits the scope of this journal and may benefit the fundamental mechanism studies and drug development for the treatment of leukemia. In general, this manuscript is well-organized and the references are updated. Key issues are required to be addressed before its publication on Cancers.

Major points:

1. In the conclusion section, regarding to the drug discovery and development, the mostly developed and the underexplored promising pathways are required to be discussed.

2. Are there any pathways/targets well-conserved and leading to significantly less drug resistance? The authors are suggested to include these contents in the manuscript.

Minor points:

1. The clinical drugs and preclinical agents in the tables are suggested to be separated or differently labeled/highlighted to improve the readability of this manuscript.

2. The summary of key biomarkers (proteins) is suggested, which would be much helpful for the identification of cell death mechanism as well as drug target identification.

Comments on the Quality of English Language

The English language is good.

Author Response

Reply to comments and suggestions for Authors

To the Editor of Cancers,

We greatly thank Editor and Referees for their interest in our manuscript entitled “Deregulation of new cell death mechanisms in Leukemia” and for reviewing it. We have carried out the appropriate changes suggested by reviewer implementing the text and tables accordingly.

Below there is a point-by-point response to referee’s questions, reporting our answers in red colour. 

We hope that the answers are satisfactory and the improvement of the manuscript makes it acceptable for publication.

Reviewer Comments:

Reviewer2 

This review summarized the regulated cell death (RCD) mechanisms in the major types of leukemia, providing researchers with a comprehensive overview of cell death and its modulation. The topic fits the scope of this journal and may benefit the fundamental mechanism studies and drug development for the treatment of leukemia. In general, this manuscript is well-organized and the references are updated. Key issues are required to be addressed before its publication on Cancers.

Major points:

Q1. In the conclusion section, regarding to the drug discovery and development, the mostly developed and the underexplored promising pathways are required to be discussed.

A1. Thanks a lot for the comment. In the conclusions section we have reported a summary of what discussed in the text. Following the questions received, we implemented the discussion about the activation of possible pathways and the molecules capable of regulating them. For some RCDs, such as necroptosis, pyroptosis or ferroptosis, several drugs are identified increasing the attractiveness of these molecular pathways. Considering the crosstalk between the RCD pathways and that for many of them, a more in-depth studies are needed, we cannot identify one as "promising" but hypothesize that all of them can be activated and can be a valid alternative to apoptotic block (page 18-19, lines 718- 740).

Q2. Are there any pathways/targets well-conserved and leading to significantly less drug resistance? The authors are suggested to include these contents in the manuscript.

A2. Thanks for the comment. Certainly, some pathways have been better characterized in leukemia, because they have been studied for longer. These obviously represent the most valid alternative blocking apoptosis to date. Among the most studied there are the pathways of necroptosis, ferroptosis and pyroptosis whose molecular interactors are well known. Since these are inflammatory deaths, it is necessary to understand better how to modulate these RCD pathways. A better characterization of them, together with all the others will better clarify the role of cell death and how it is possible to induce it specifically in leukemic clones.  (Page 3 lines 97-106; conclusion section)

Minor points:

Q1. The clinical drugs and preclinical agents in the tables are suggested to be separated or differently labeled/highlighted to improve the readability of this manuscript.

A1 Thanks a lot for the suggestion. Preclinical studies have been formulated for all the compounds shown in the tables. Only two are in clinical trials. To make the tables more readable, the clinical trial studies have been highlighted for these two drugs only. Both are discussed further in the text (page 13, lines 503-508) (pages 13-14, lines 535-540). The corresponding tables are also adapted (page 13, Table 3) (page 14, Table 4).

Q2. The summary of key biomarkers (proteins) is suggested, which would be much helpful for the identification of cell death mechanism as well as drug target identification.

A2 Thanks for the comment. We have added a table where we have reported the key biomarkers and their related cell death pathway (page 4, Table 1).

Reviewer 3 Report

Comments and Suggestions for Authors

In this review, Favale and colleagues provide a comprehensive description of the advances made in the field of the molecular mechanisms that drive new regulated cell death (RCD) processes and their deregulation in different types of leukemia. In addition, the authors provide an overview of known or newly characterized pharmacological strategies capable of modulating the specific mechanisms of RCD. The manuscript is well written, the topics are well stated and supported by appropriate references and I think it could be published in its present form.

I only have minor points:

Lines 46-47: Remove one of the "and" in the sentence.

Line 155: IL-1b should be IL-1β, also in other part of the manuscript.

Line 308: Reference 83, maybe you mean ref 84.

Line 356: Remove PMID:31719677.

Line 363: Reference 122 should be 102.

Line 371: specify MPO acronym meaning.

Line 468:  TNF-α

Line 469: specify z-VAD acronym meaning.

Comments on the Quality of English Language

Minor editing of English language required.

Author Response

Reply to comments and suggestions for Authors

To the Editor of Cancers,

We greatly thank Editor and Referees for their interest in our manuscript entitled “Deregulation of new cell death mechanisms in Leukemia” and for reviewing it. We have carried out the appropriate changes suggested by reviewer implementing the text and tables accordingly.

Below there is a point-by-point response to referee’s questions, reporting our answers in red colour. 

We hope that the answers are satisfactory and the improvement of the manuscript makes it acceptable for publication.

Reviewer Comments:

Reviewer3

In this review, Favale and colleagues provide a comprehensive description of the advances made in the field of the molecular mechanisms that drive new regulated cell death (RCD) processes and their deregulation in different types of leukemia. In addition, the authors provide an overview of known or newly characterized pharmacological strategies capable of modulating the specific mechanisms of RCD. The manuscript is well written, the topics are well stated and supported by appropriate references and I think it could be published in its present form.

Q1. I only have minor points.

A1 We thank you for your comment and apologize for these errors. Following your comment, we have corrected the text accordingly. The corrections are all indicated in red in the updated text.

Lines 46-47: Remove one of the "and" in the sentence.

Line 155: IL-1b should be IL-1β, also in other part of the manuscript.

Line 308: Reference 83, maybe you mean ref 84.

Line 356: Remove PMID:31719677.

Line 363: Reference 122 should be 102.

Line 371: specify MPO acronym meaning.

Line 468:  TNF-α

Line 469: specify z-VAD acronym meaning.
